# The effects of geographical distributions of buildings and roads on the spatiotemporal spread of canine rabies: An individual-based modeling study

Chayanin Sararat[1], Suttikiat Changruenngam[1], Arun Chumkaeo[2], Anuwat Wiratsudakul[3], Wirichada Pan-ngum[4], Charin Modchang[1,5]*

1 Biophysics Group, Department of Physics, Faculty of Science, Mahidol University, Bangkok, Thailand, 2 Songkhla Provincial Livestock Office, Muang, Songkhla, Thailand, 3 Department of Clinical Sciences and Public Health, and the Monitoring and Surveillance Center for Zoonotic Diseases in Wildlife and Exotic Animals, Faculty of Veterinary Science, Mahidol University, Nakhon Pathom, Thailand, 4 Department of Tropical Hygiene, Faculty of Tropical Medicine, Mahidol University, Bangkok, Thailand, 5 Centre of Excellence in Mathematics, CHE, Ministry of Education, Bangkok, Thailand

* charin.mod@mahidol.edu

**Data Availability Statement:** All relevant data are within the manuscript and its Supporting Information files.

## Abstract

Rabies is a fatal disease that has been a serious health concern, especially in developing countries. Although rabies is preventable by vaccination, the spread still occurs sporadically in many countries, including Thailand. Geographical structures, habitats, and behaviors of host populations are essential factors that may result in an enormous impact on the mechanism of propagation and persistence of the disease. To investigate the role of geographical structures on the transmission dynamics of canine rabies, we developed a stochastic individual-based model that integrates the exact configuration of buildings and roads. In our model, the spatial distribution of dogs was estimated based on the distribution of buildings, with roads considered to facilitate dog movement. Two contrasting areas with high- and low-risk of rabies transmission in Thailand, namely, Hatyai and Tepha districts, were chosen as study sites. Our modeling results indicated that the distinct geographical structures of buildings and roads in Hatyai and Tepha could contribute to the difference in the rabies transmission dynamics in these two areas. The high density of buildings and roads in Hatyai could facilitate more rabies transmission. We also investigated the impacts of rabies intervention, including reducing the dog population, restricting owned dog movement, and dog vaccination on the spread of canine rabies in these two areas. We found that reducing the dog population alone might not be sufficient for preventing rabies transmission in the high-risk area. Owned dog confinement could reduce more the likelihood of rabies transmission. Finally, a higher vaccination coverage may be required for controlling rabies transmission in the high-risk area compared to the low-risk area.

**Funding:** AW was funded by the National Science and Technology Development Agency (NSTDA), Thailand (Grant ID. P-18-51758) https://www.nstda.or.th/en/. CS is supported by the Science Achievement Scholarship of Thailand (SAST) https://science.mahidol.ac.th/th/scholarships.php. The funders had no role in study design, data collection and analysis, decision to publish, or preparation of the manuscript.

**Competing interests:** The authors have declared that no competing interests exist.

## Author summary

Canine rabies is responsible for tens of thousands of human deaths annually worldwide, primarily in Asia and Africa. In Thailand, a sharp increasing trend of animal rabies cases was recently observed during 2014 and 2018, in which the numbers of cases were 250 and 1,105, respectively. As a directly transmitted disease, geographical distributions of buildings where dogs can live in, and road networks could inevitably influence the spatiotemporal spread of rabies. To investigate the role of these geographical structures on the transmission dynamics of canine rabies, we developed a stochastic individual-based model that integrates the exact geographical distributions of buildings and roads. The spatial distribution of dogs was assumed according to the distribution of buildings, with roads included to help dogs travel around. The model was then applied to investigate rabies transmission in the low-risk and high-risk areas in Thailand. Our modeling results highlighted that only differences in the geographical structures of buildings and roads could result in the difference in rabies transmission dynamics in these two areas. We also explored the impacts of reducing dog population, restriction of owned dog movement, and dog vaccination on the rabies spread. We discovered that lowering the dog population alone might not be enough to keep rabies from spreading in the high-risk area. Owned dog confinement may lower the risk of rabies transmission even more. In addition, different levels of vaccination coverages are probably required to control rabies in different geographical settings.

## Introduction

Rabies is a fatal zoonotic disease caused by a *Lyssavirus* belonging to the family Rhabdovirudae [1]. This virus is responsible for around 59,000 human deaths annually worldwide [2]. Dogs have been identified as the main reservoirs for the rabies virus in many developing countries, mostly located in Asia and Africa, including Thailand [2]. In 2015, there was a call for global efforts to set a goal for zero human dog-mediated rabies deaths by 2030 worldwide [3]. To achieve this, the prevention and control policies for rabies issued by affected countries must be driven by scientific evidence and implemented effectively. The factors that greatly contribute to the spread of the virus must be revealed, and the policies should be revised accordingly.

As dogs serve as the main source of human rabies, preventing and controlling canine rabies will inevitably prevent human rabies. High-coverage dog vaccination has been widely recommended to eliminate canine rabies. The World Health Organization (WHO) suggested that an effective vaccination program should reach at least 70% of the population [4]. In various contexts, however, different vaccination coverages, as well as vaccination strategies, may be required. Dog sterilization is an approach for controlling the reproduction of the dog population, specifically to stabilize or reduce the population size. In India, for example, using this method in conjunction with dog vaccination proved beneficial in controlling rabies [5]. Alternatively, dog culling has been practiced, even though this approach was unsuccessful in several areas [6–8] In China, dog vaccination programs alone might be insufficient to control rabies; however, dog culling in combination with vaccination has been recommended to enhance the rabies control [9].

In Thailand, the number of human rabies cases was as high as 370 cases in 1980 [10]. With the great attempts over decades, the number had gradually declined to only three cases in 2020 [11]. However, the rabies situation in animals is far worse than in humans these days. A sharp increasing trend of animal rabies cases was recently observed during 2014 and 2018, in which

the numbers of cases were 250 and 1,105, respectively. Unsurprisingly, the majority of the cases were dogs [12]. As these animals are the major reservoirs for rabies, rabies transmission to humans by dog bites may happen anytime, and the number of human rabies cases in Thailand may rise again.

Mathematical modeling has become a tool for investigating the rabies transmission dynamics and exploring rabies intervention strategies [13–18]. Rabies transmission models are usually based on a compartmental structure in which the host population is classified into different groups according to their epidemiological status [18–20]. However, due to the localized nature of the rabies transmission process, incorporating the spatial distribution of hosts into an epidemic model is usually necessary for more accurate model prediction. In this regard, a spatial-dependent diffusion process was integrated into previous rabies transmission models both in wildlife populations [21–23] and dog populations [24]. In these models, the dispersal of animals was modeled using a spatial-dependent diffusion process. Metapopulation-type models were also used to investigate the spatial spread of rabies [25–28]. For this approach, the spatial transmission of rabies was represented by the transmission between spatial patches. Besides, empirical data such as individual positions and contact patterns among dogs have also been incorporated into epidemic models to investigate the spatiotemporal propagation of rabies [29–32].

Geographical features are important determinants of distribution, density, and movement of host populations [17,33–35]. For example, natural barriers such as rivers and mountains can halt while roads can facilitate host movement and could affect the rabies transmission dynamics [13,17,33,36,37]. Therefore, the geographical features have been purposely combined with rabies transmission models. By using a stochastic spatial model including alignment of rivers, Smith et al. found that rivers could reduce the speed of raccoon rabies spread by approximately 7-fold [27]. Neilan and Lenhart also considered impacts of the heterogeneous spatial domain, including a river and forest, on the spread of the disease in wild raccoons [22].

In this study, we constructed a stochastic individual-based model that integrates the exact geographical locations of buildings and alignments of roads. The spatial distribution of dogs in the model was estimated according to the distribution of buildings, and roads were assumed to facilitate dog movement. The constructed model was then employed to investigate the geotemporal transmission dynamics of canine rabies in two contrasting areas with high and low risk of rabies transmission in Thailand. Finally, the influence of rabies interventions such as dog vaccination, dog population reduction, and owned dog movement restrictions on the spread of canine rabies was explored.

## Materials and methods

### Study sites

Songkla is a province that has the highest risk of rabies transmission in the southern part of Thailand. In this work, two districts in Songkla with a contrasting risk of rabies occurrence, namely, Hatyai and Tepha, were chosen as case studies. Hatyai was classified as a high-risk district in Songkla, reporting approximately 8.8 rabid dogs per year during the years 2016–2020. In contrast, Tepha was classified as a low-risk district that reports no rabid dog in the past five years.

### Geographical maps

The geographical maps aggregating layers of administrative boundaries, polygons representing buildings, and polyline of roads were retrieved from the Department of Public Works and

Town & Country Planning, Ministry of Interior, Thailand. The buildings were categorized into three groups according to where dogs usually reside. The first group (G1) contains houses and residential buildings. Owned dogs were assumed to be found only in G1 buildings. However, since some G1 buildings have no confinable fences, dogs living in these non-confined buildings can roam freely. In our model, G1 buildings were randomly chosen to be enclosed by fences (Table 1). The second group (G2) involves public places where unowned free-roaming dogs can usually be found as there are ample food and shelters. This group of buildings comprises temples, schools, and fresh markets. Finally, the third group (G3) holds other types of buildings, including groceries, hotels, banks, etc. Some unowned free-roaming dogs live near these buildings as people sometimes feed them with some foods. The geographical location of each building was represented by the centroid of the building. However, if a public place, e.g., a school or a temple, has more than one building, its geographical location was represented by the centroid of all the buildings of that public place.

**Table 1. Parameters and the baseline values used in the rabies transmission model.**

| Parameters | Values | Details and references |
|---|---|---|
| Natural birth rate ($b$) | $2.28\times10^{-4}$ day$^{-1}$ | [41] |
| Mortality rate ($m$) | $m = bS/N$ | - |
| Importation rate ($\varepsilon$) | 0.0082 day$^{-1}$ | Estimated based on incidence data |
| Latency rate ($\sigma$) | 0.0448 day$^{-1}$ | [42] |
| Rabies induced death rate ($\delta$) | 0.32 day$^{-1}$ | [42] |
| Vaccination coverage | 50.38% | Survey report |
| Model parameter representing the likelihood that a susceptible dog will be infected by an infectious dog ($p$) | $1\times10^{-5}$ | Model fitting with R-square = 0.9772 |
| Mean travelling distance of a susceptible dog | | Assumed, with sensitivity analysis shown in S2 and S3 Figs |
| - living in a building near roads with confinable fences | 0.1 km | |
| - living in a building near roads without confinable fences | 1 km | |
| - living in a building far from roads with confinable fences | 0.05 km | |
| - living in a building far from roads without confinable fences | 0.5 km | |
| Mean travelling distance of a rabid dog | | Assumed, with sensitivity analysis shown in S2 and S3 Figs |
| - living in a building near roads with confinable fences | 0.1 km | |
| - living in a building near roads without confinable fences | 5 km | |
| - living in a building far from roads with confinable fences | 0.05 km | |
| - living in a building far from roads without confinable fences | 2.5 km | |
| Proportion of household with fence | 0.48 | [43] |
| Proportion of G1 buildings that own dogs | 0.54 | [44] |
| Proportion of G2 buildings that own dogs | 0.9 | Assumed, with sensitivity analysis shown in S4 Fig |
| Proportion of G3 buildings that own dogs | 0.1 | Assumed, with sensitivity analysis shown in S5 Fig |
| Average number of dogs per dog-owning household | 2.67 | [43] |
| Average number of dogs per G2 building | 10 | Assumed, with sensitivity analysis shown in S6 Fig |
| Average number of dogs per G3 building | 1 | Assumed, with sensitivity analysis shown in S7 Fig |

We also classified a building as a near-to-road building or a far-from-road building based on its shortest building-to-road distance. To estimate the shortest building-to-road distance of each building, we created local points along the roads with the spacing distance between local points of 5 meters. The shortest building-to-road distance was estimated as the shortest distance between the centroid of the building and the shortest local point on the road. A building with the shortest building-to-road distance less than the corresponding median distance was classified as a near-to-road building. Building centroid coordinates and locations along roadways were extracted using QGIS software (version 3.16.10). All figures depicting the geographical distribution of the man-made structures as well as the spatiotemporal spreading patterns of the disease were created using MATLAB software (version R2020b, The MathWorks, Inc).

## Population and spatial distribution of dogs

In our model, dogs are classified into three types: owned dogs, owned free-roaming dogs, and unowned free-roaming dogs. Owned dogs were assumed to be found in G1 buildings that have confinable fences, whereas owned free-roaming dogs could be found in G1 buildings with no confinable fences. The number of owned dogs was calculated by multiplying the number of G1 buildings with fences, the proportion of G1 buildings owning dogs, and the average number of owned dogs per household. Similarly, the number of owned free-roaming dogs was estimated by multiplying the number of G1 buildings with no fences, the proportion of G1 buildings owning dogs, and the average number of owned dogs per household.

For unowned free-roaming dogs, although they were ownerless, they are usually fed by local feeders, e.g., Buddhist monks and kind aunties [26]. Therefore, these dogs live near public places (G2 or G3 buildings) where local feeders usually provide them food. Since G2 and G3 buildings have no confinable fences, these unowned dogs can roam freely. The numbers of unowned free-roaming dogs residing in G2 and G3 buildings were estimated based on the number of buildings in each category, the proportion of buildings that own dogs, and the average number of unowned free-roaming dogs per dog-owning buildings (Table 1). Based on our assumption, about 91% of unowned free-roaming dogs in Hatyai and 99% of unowned free-roaming dogs in Tepha live in G2 buildings, while the rest live in G3 buildings (Fig 1A).

## Model structure

We constructed a spatially explicit individual-based model of dog rabies transmission. The model integrates the exact configurations and geographical locations of dogs, buildings, and roads. Dogs are also classified as owned, owned free-roaming, or unowned free-roaming dogs based on the type of building that they live. The model classifies dogs into the following four epidemiological classes: susceptible ($S$), exposed ($E$), infectious ($I$), and vaccinated ($V$) (Fig 1B). A susceptible dog can be infected, through a bite of an infectious dog, under the force of infection $\lambda$. After being infected, the susceptible dog progresses to the exposed class. Dogs in this class have already acquired the infection but are not yet infectious and cannot transmit the virus to other susceptible dogs. Exposed dogs become infectious at a rate $\sigma$, which is inversely proportional to the latent period. Infectious rabid dogs eventually die at a rate $\delta$. All newborn dogs are assumed to be susceptible to the disease and enter the susceptible class at a rate $b$. To maintain a constant size of the dog population, dogs in all classes were assumed to naturally die at a mortality rate $m = bS(t)/N(t)$. Susceptible dogs, both owned and unowned, are vaccinated at a rate $v$, while vaccinated dogs lost their vaccination immunity at a rate $\omega$. In order to conserve vaccination coverage of the population, the vaccination rate was set to $v = (\omega+m)N_S(t)/N_V(t)$, where $N_S(t)$ and $N_V(t)$ is the total number of susceptible and vaccinated dogs at time $t$, respectively. In addition to the local transmission, the model also considers the

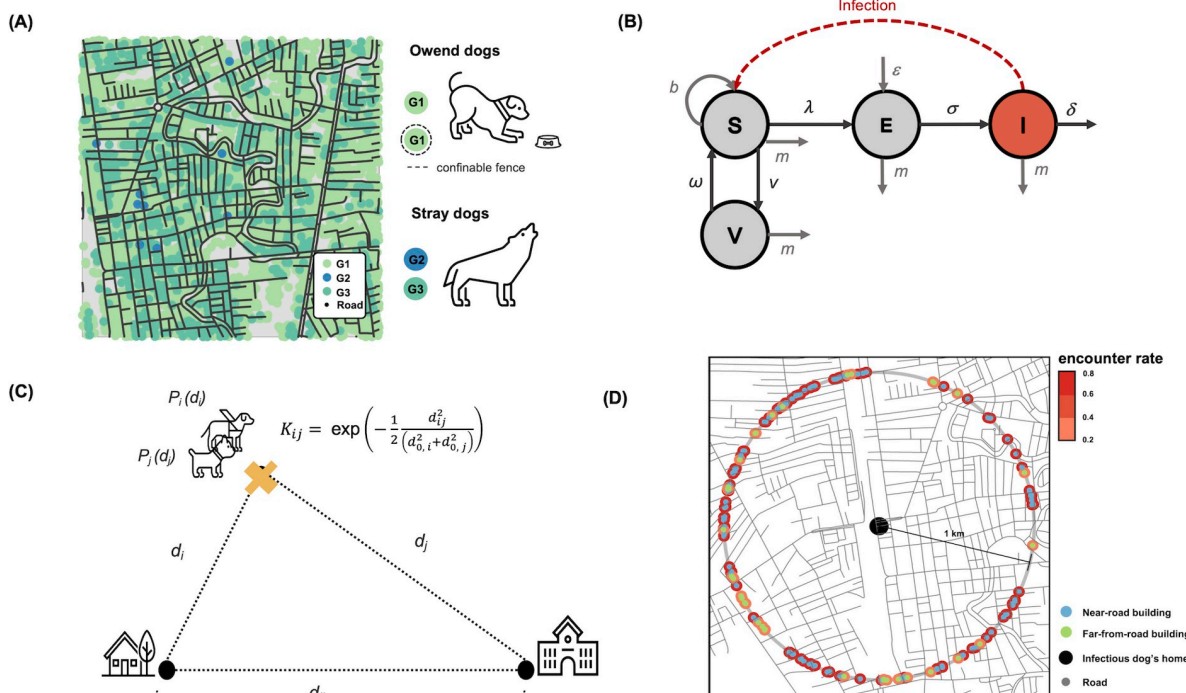

**Fig 1. Structure of the canine rabies transmission model.** (A) Example of the geographical distribution of buildings and roads in an area of 2x2 km$^2$. Owned dogs reside only in G1 buildings, while unowned free-roaming dogs may occupy either G2 or G3 buildings. (B) Schematic of the transmission model. Based on the infection status, the model classifies each dog into susceptible ($S$), exposed ($E$), infectious ($I$), and vaccinated ($V$) classes. The dashed arrow represents transmission events, and the solid arrows indicate transitions between compartments. (C) Illustration of the probability of finding a dog at distance $d$ from their home location, $P(d)$, and the unnormalized encountering rate between dog $i$ and dog $j$, $K_{ij}$. (D) An illustrative example of the unnormalized encountering rate between a rabid dog residing at the centered black point and susceptible dogs living one kilometer apart. The blue and green dots on the map represent the home locations of susceptible dogs that are located near roads and far from roads, respectively. The unnormalized encountering rates are indicated by the colors of the dot circumferences.

importation of latently infected dogs from surrounding areas, which can occur at a rate $\varepsilon$. The importation rate was estimated from the incident data where a rabid dog that was identified after 25.4 days (i.e., sum of the latent period and infectious period) since the previous most recent reported rapid dog was assumed to be an imported case. All imported cases were assumed to be unowned free-roaming dogs.

An (unnormalized) encountering rate of dog $i$ and dog $j$ staying a distance $d_{ij}$ apart ($K_{ij}$) can be written as (see S1 Text for the derivation)

$$K_{ij} = \exp\left(-\frac{1}{2}\frac{d_{ij}^2}{(d_{0,i}^2 + d_{0,j}^2)}\right),$$

where $d_{0,i}$ and $d_{0,j}$ denotes the mean traveling distance of dog $i$ and dog $j$, respectively (Fig 1C and 1D). The force of infection of susceptible dog $i$ at time $t$ can be written as $\lambda_i(t) = \sum_{j=1}^{N_I(t)} pK_{ij}$, where $p$ is a parameter representing the likelihood that a susceptible dog will be infected by an infectious dog, and $N_I(t)$ is the total number of infectious dogs at time $t$. To estimate the value of p, a computational search algorithm was employed [38]. Specifically, the model searched for the value of $p$ that provides the best fit between the simulation result and the reported cumulative rabies cases. Possible values of $p$ that the model can search for are uniformly distributed in the interval $[10^{-6}, 10^{-4}]$ with $10^{-6}$ resolution.

Dog roaming behaviors might be different in various environments. Free-ranging domestic dogs in rural areas in Chad, for example, have home ranges that correspond to a few kilometers of a circular radius [31], whereas free-roaming domestic dogs in Chad, Guatemala, Indonesia, and Uganda move little more than one hundred meters from their home [39]. In the lack of empirical data of dog movement in our study sites, we assumed that free-roaming dogs, i.e., dogs residing in buildings without confinable fences, travel at a mean distance of 500 meters from their homes. Confined owned dogs, i.e., dogs living in the G1 buildings with confinable fences, were assumed to travel around their home territory with a mean distance of 50 meters. Our model assumed that roads would facilitate dog wandering [13,17,33,36,37]; thus, dogs living in buildings near roads could travel further than dogs far from roads. (Table 1). However, a sensitivity analysis of the dog's traveling distance was also performed by scaling the mean traveling distances by factors of 0.5 and 2. Note also that although there is a study pointed out that high-traffic roads could be major geographical obstacles to dog roaming [25], the road networks in our study areas are dominated by small local roads (96.65% for Hatyai and 94.59% for Tepha, S1 Fig), we hence did not consider the possible barrier role of roads in our study.

We employed the Gillespie algorithm to simulate the model stochastically [40]. At the starting time, there was only one infectious individual whose location was randomly assigned to a particular building, while other dogs were susceptible or vaccinated dogs, depending on the initial vaccination coverage. The model simulations were implemented using MATLAB R2020b. The parameters used in the model are summarized in Table 1.

## Results

### Geographical characteristics of Hatyai and Tepha districts

In this study, the rabies simulations were based on data from two distinct regions in the southern part of Thailand: Hatyai district and Tepha district (Fig 2A). Both Hatyai and Tepha districts are located in the Songkla province. Hatyai is the largest metropolitan area in the Songkla province, with an area of 853 km$^2$. According to the geographical map provided by the Department of Public Works and Town & Country Planning, Ministry of Interior, Thailand, there are 152,439 buildings in Hatyai. Among these 152,439 buildings, 144,532 (94.81%) are group-1 (G1) buildings (e.g., houses, residences, and other small buildings), 887 (0.58%) are group-2 (G2) buildings (e.g., schools, temples, markets), and 7,020 (4.61%) are group-3 (G3) buildings (e.g., groceries, hotels, banks) (Fig 2B and S1 Table). The Tepha district covers an area of 978 km$^2$. Although Tepha has a larger geographical area than Hatyai, it comprises only 33,652 buildings (Fig 2C and S2 Table). Of these, 33,275 (98.88%) are G1 buildings, 75 (0.22%) are G2 buildings, and 302 (0.90%) are G3 buildings. In addition, based on the number of buildings, the estimated number of dogs in Hatyai was 207,818 and it was 48,142 in Tepha.

Spatial analysis of the distribution of buildings in these two districts was also performed. We found that these two districts have marked differences in the spatial distributions of buildings (Fig 3). In Hatyai, the buildings are densely agglomerated in the downtown area, while, in Tepha, the buildings are more evenly distributed throughout the district. We also measured the pairwise distances between a given building and all other buildings and sorted the pairwise distances from shortest to longest. The median pairwise distance with a different rank of closeness (from shortest to longest) is shown in Fig 3C. We found that the buildings in Hatyai are generally located closer together than the buildings in Tepha. In addition, the difference in the median pairwise distance in these two districts was more pronounced when a higher rank of closeness was considered (Fig 3C).

As roads have been found to facilitate the mobility of dogs [33,36,37], we also measured the shortest building-to-road distance of all buildings in these two districts. We found that the

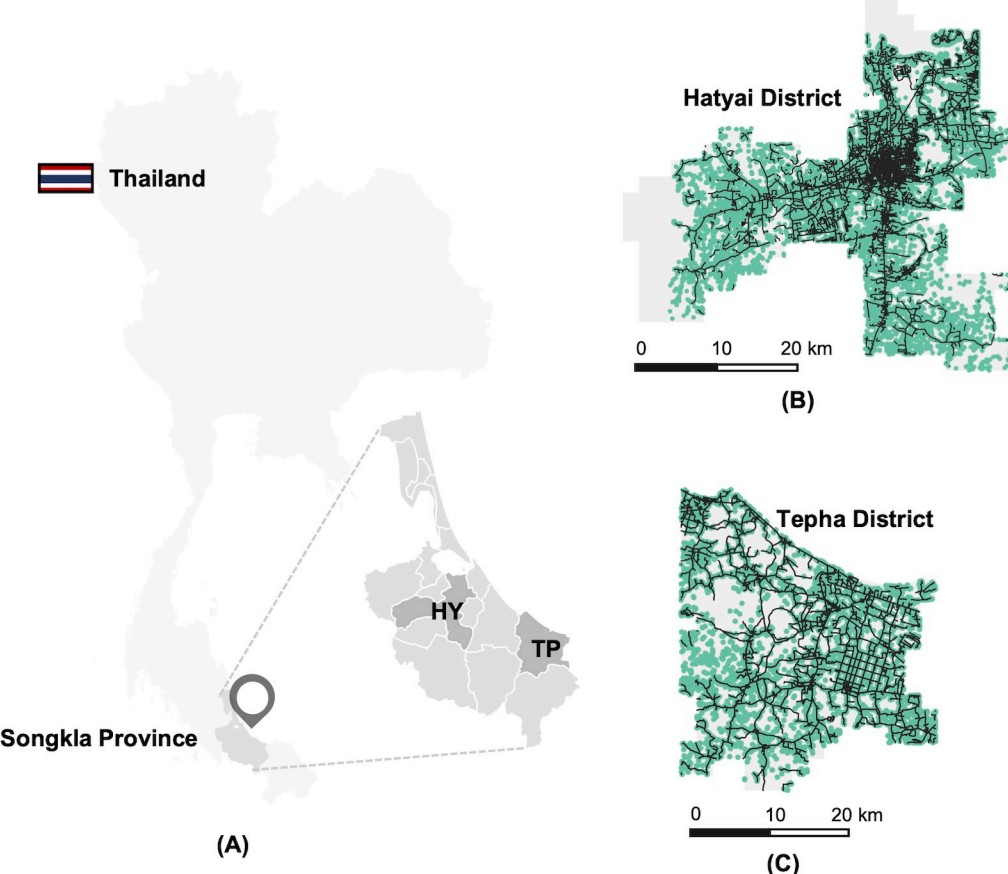

**Fig 2. Study sites and geographical distribution of buildings and roads.** (A) The locations of the study sites, Hatyai and Tepha district, in Thailand. (B and C) Spatial distribution of buildings (green patches) and roads (black lines) across Hatyai and Tepha district. The base layer of the map was obtained from https://data.humdata.org/dataset/thailand-administrative-boundaries.

buildings in Tepha have generally located farther away from the roads as compared to the buildings in Hatyai (Fig 3D). The median and mean of the shortest building-to-road distance in Hatyai were 24 meters and 53 meters, respectively, while in Tepha, the corresponding values were 30 meters and 101 meters, respectively. Although the medians of the shortest building-to-road distance in these two districts were fairly close, the mean of the shortest building-to-road distance in Tepha was almost twice the corresponding mean in Hatyai. This indicated that the spatial distribution of buildings in Tepha is more dispersed than in Hatyai.

### Influence of geographical characteristics on the spatiotemporal transmission dynamics of rabies

To investigate how different geographical distribution of buildings and roads could affect the spatiotemporal transmission dynamics of rabies, we separately simulated our rabies transmission model in Hatyai and Tepha using the same set of model parameters. We found that our model simulations can capture rabies transmission dynamics in Hatyai well (Fig 4A). In Tepha, where rabies had been undetected for more than five years, the model prediction illustrated that there would be a very low chance for an outbreak in this district when an infected dog was imported (averaged cumulative local cases were fewer than one (Fig 4B). We also

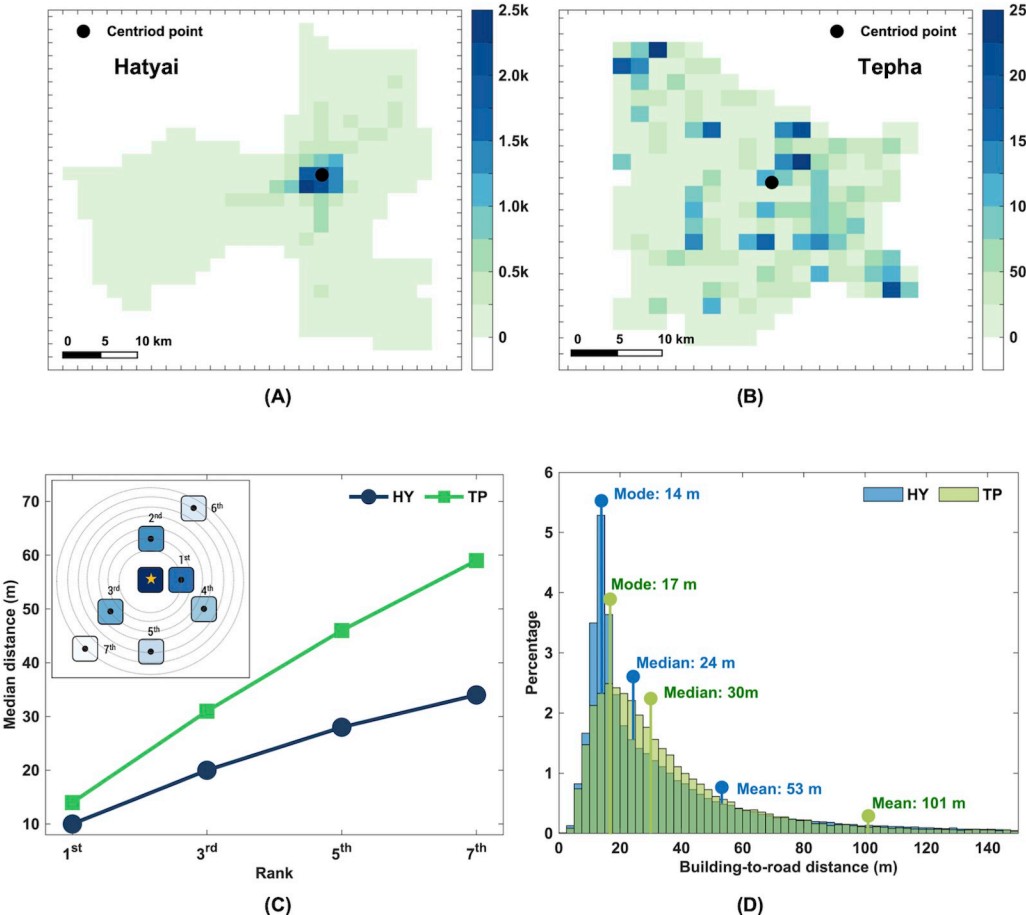

**Fig 3. Building density and distribution.** (A and B) Spatial distribution of building density in Hatyai and Tepha. The color bar indicates the building density in the unit of buildings per square kilometer. Note that the color bar in Hatyai represents 10 times higher building density than in Tepha. The black circles show the centroid points of the building distribution. (C) The median pairwise distance between two buildings with a different rank of closeness (from shortest to longest) in Hatyai (HY) and Tepha (TP) district. The inset illustrates an example of a building arrangement with different ranks of closeness relative to the stared building. (D) The distribution of the shortest building-to-road distance in Hatyai (HY) and Tepha (TP).

estimated the likelihood of an imported rabid dog causing a secondary infection. It was found that a randomly imported dog in Hatyai and Tepha has the likelihood of infecting another susceptible dog of 0.18 and 0.03, respectively. A sensitivity analysis has also been performed when the imported dog is owned rather than unowned. In this case, we found that the likelihood of secondary infection due to the imported owned dog in Hatyai and Tepha is 0.18 and 0.02, respectively. This result indicated that the chance of rabies extinction in Tepha would be approximately six to nine times higher than that in Hatyai. In addition, as the exact traveling distances of dogs are not known, we also perform the sensitivity analysis of the traveling distance. We found that when the distances traveled by dogs increase, the numbers of cumulative cases and the likelihood of making a secondary infection also increase, and vice versa (S2 Fig).

A geo-temporal spreading pattern of rabies in Hatyai and Tepha is shown in Fig 5. At the beginning of the simulation, there was only one rabid dog at a randomly chosen location. We found that at the early time of the outbreak, rabid dogs were more likely to be found in areas with high building densities and then spread to areas with lower building densities. However, the rabies transmission pattern in Tepha seems more dispersed than in Hatyai, which might be

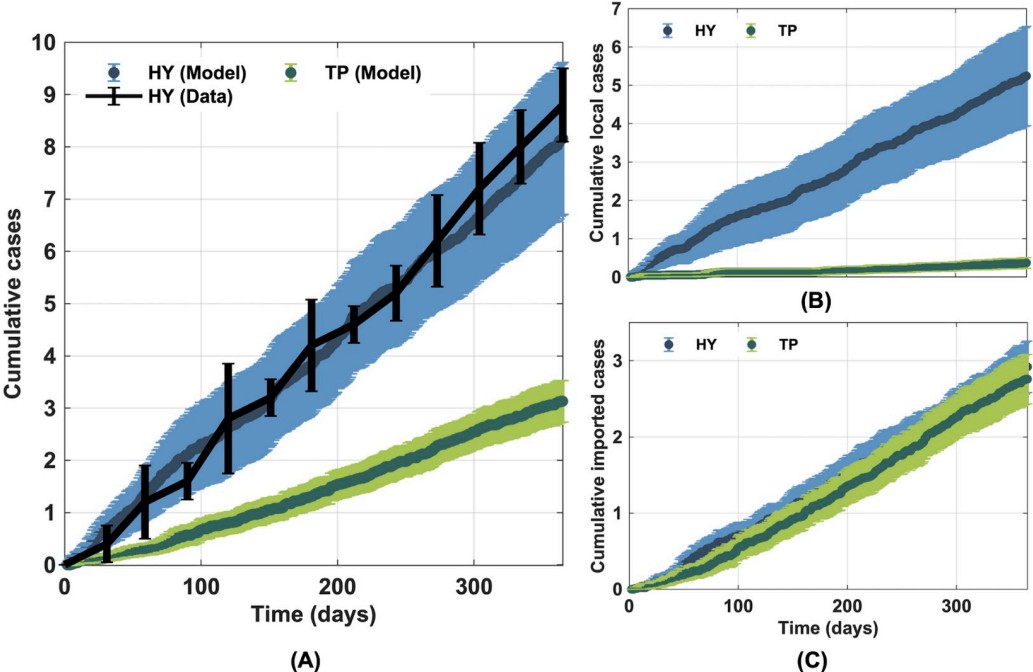

**Fig 4. Effect of the geographical distribution of buildings and roads on the rabies transmission dynamics.** (A) Comparison of the cumulative number of locally transmitted rabid dogs obtained from the simulations cases simulated in both Hatyai and Tepha and the five-year averaged reported data from Hatyai (years 2016–2020). (B) The number of cumulative locally transmitted cases, and (C) the number of cumulative imported cases in Hatyai (HY) and Tepha (TP).

due to the fact that the building distribution in Tepha has more dispersion than in Hatyai. A sensitivity analysis of the traveling distance was also performed. We found that when the distances traveled by dogs increase, the spreading pattern of rabies in Hatyai was found to be more dispersed, while the spreading pattern in Tepha is generally not changed (S3 Fig).

## Effect of canine rabies interventions

The proposed model allows us to investigate the effect of different intervention strategies (i.e., reducing the dog population, confinement of owned dogs, and mass vaccination) on the likelihood of an imported infected dog to make a secondary infection, which was estimated as a ratio of the number of the model realizations that contain at least one locally infected dog. For the baseline scenario, the estimated number of dogs, based on the number of buildings, was 207,818 in Hatyai and 48,142 in Tepha. Reducing dog population size influences the density of the susceptible dogs, which in turn might also affect the likelihood of disease transmission. As shown in Fig 6A, we found that reducing the dog population size can only slightly decrease the transmission likelihood if the reduction level is not high enough. When 20% of the whole population is removed from Tepha, the likelihood drops by just 7% (from 0.030 to 0.028), but when 60% and 80% of the population are removed, the chance drops by 67% and 74%, respectively. In Hatyai, eliminating 20% of the population reduces the likelihood of rabies transmission from an imported rabid dog by just 8%. Moreover, even after reducing by 80%, the likelihood was diminished by only 50%. The same tendency was also found for different dog-traveling ranges; however, the longer dogs can go, the greater the chance of transmission. In addition, even if the number of unowned free-roaming dogs was specifically reduced by lowering the fraction of buildings owning unowned dogs or the average number of unowned free-

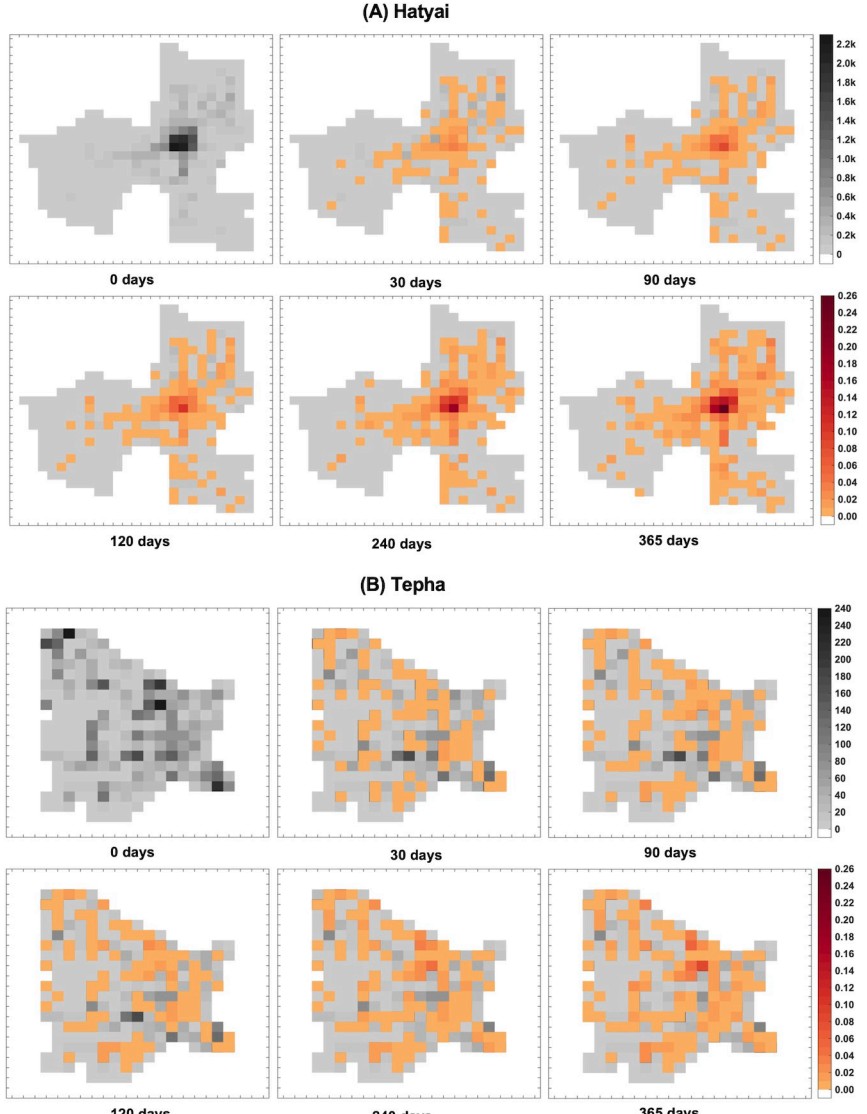

**Fig 5. Geotemporal spreading pattern of canine rabies in Hatyai and Tepha.** The snapshots showing the spreading patterns of rabies, averaged from 100 simulations, in Hatyai (A) and Tepha (B) at 0, 30, 90, 120, 240, and 365 days after the introduction of an index rabid-dog. The greyscale indicates the density of buildings (buildings/km$^2$), and the warm-color scale indicates the density of the cumulative number of rabid dogs (dogs/ km$^2$) in each cell.

roaming dogs per building owing unowned dogs, this way did not have much impact on the likelihood, as depicted in S4 and S5 Figs. These results indicated that only reducing the dog population without any other complementary measures might not be adequate to eliminate rabies. However, reducing the dog population could lower the size of the outbreak in dogs and, therefore, reduce the potential transmission going into humans.

We also examined the potential of confinement of owned dogs in halting the transmission of rabies. We found that as the proportion of dogs that live in the confined houses increased, the likelihood for an imported rabid dog to make a secondary infection was greatly reduced (Fig 6B). The likelihood of secondary infection in Hatyai fell from 0.33 to 0.28 (15%) after a 20% restriction, whereas the relevant risk in Tepha went down from 0.06 to 0.04 (33%). The likelihood was decreased by 62% and 60% in Hatyai and Tepha, respectively, when 60% of the

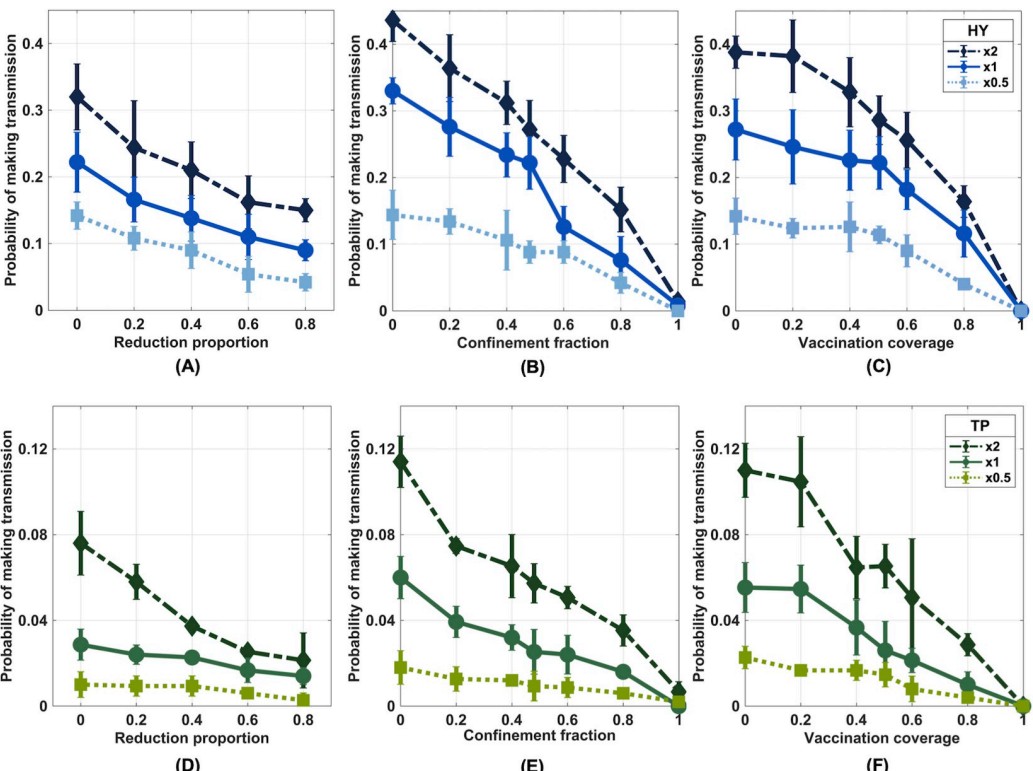

**Fig 6. Impact of intervention strategies on the likelihood of rabies transmission in Hatyai (A, B, and C) and Tepha (D, E, and F).** The likelihood of rabies transmission caused by an imported infected dog under the intervention scenarios of (A and D) reducing the dog population size, (B and E) reducing the fraction of free-roaming dogs, and (C and F) increasing the mass vaccination coverage. A sensitivity analysis where the dog traveling distances are scaled by factors of 0.5 (x0.5) and 2 (x2) was also shown (dashed lines).

owned dogs were confined. We also found that the traveling ranges of dogs have greater influences on the transmission risk when the confinement fraction is small.

In order to achieve dog-mediated human rabies elimination, mass dog vaccination is an approach recommended by the World Health Organization (WHO). As demonstrated in Fig 6C, when both owned and unowned dog populations were randomly vaccinated, the likelihood of an imported infected dog causing a secondary infection was lower with the increase of vaccination coverage and eventually disappeared when the whole population was immunized. For instance, in Tepha, vaccinating 80% of the dog population can mitigate the likelihood by 82%, and in Hatyai, it cut the risk by 57%. Consistently, when more dogs are protected by the vaccine, the distances dogs can travel less affects the probability of transmission. However, shorter dog-travel distances help reduce the likelihood of disease transmission in places where vaccination coverage is low.

## Discussion

Although several modeling studies have investigated the influence of geographical features, such as rivers, roads, and buildings, on rabies spreading [7,17,19,33,35], none of these focused on the small scale of the spatial distribution of these features. Therefore, in this work, we constructed an individual-based epidemic model for canine rabies transmission incorporating the exact geographical distribution of buildings and road networks. We derived the encounter rate of a rabid dog and a susceptible dog that live at different locations based on the assumption of

the random movement of rabid animals [20–22]. As evidence indicated that importation of infected dogs might be a cause for the persistence of rabies in a disease endemic region [7,25], rabies importation was also considered in our model.

Even though Songkla province exhibited the highest risk of rabies occurrence, the risk varied substantially across districts in the province. With a curiosity to know what brings about the heterogeneity in the risk of disease occurrence, two contrast districts with one high-risk and the other low risk of the disease occurrence were selected in this study. Our analysis clearly revealed the difference in the spatial pattern of the geographical feature distributions in these two sites. Buildings in the Hatyai district tend to clump together, and the density of buildings declines sharply from the core to the border (Fig 3). In contrast, the buildings in Tepha seem to be randomly distributed, and there is no clear pattern of the building density. We hypothesized that the difference in the distribution of buildings and roads in these two districts might contribute to the existence of rabies.

To test the hypothesis, we simulated and compared the rabies transmission dynamics in Hatyai and Tepha districts. We found that under the same set of epidemiological parameters, the disease incidence rate in Hatyai is higher than in Tepha. Therefore, our simulation results indicated that the difference in the geographical feature distribution in these two districts might be one of the main factors contributing to the difference in the rabies transmission dynamics in these two areas.

Since the rabies virus is transmitted through direct contact, usually from a deep bite or scratch of rabid dogs, the disease can be mostly transmitted to nearby susceptible dogs within the traveling range of the rabid dogs [19]. In order to spread out the disease, the transmission chain may consist of several spatial transmission segments. The presence of unoccupied or sparely occupied areas could cut the transmission chain, thus halting the spatial spread of the disease spread. In contrast, the continuous arrangements of the inhibiting buildings could facilitate the disease spreading. In addition, a very high-density core area could also serve as a sanctuary region where a chain of transmission can sustain (e.g., rabies transmission in Hatyai (Fig 3A). Our work also highlighted a correlation between the density of residential buildings and the rabies incidences (Fig 5). Like other directly transmitted diseases, local dog population density is a key determinant of rabies transmission [7,29,33,42,45–47]. In the crowded area, there are more susceptible dogs within a traveling distance of a rabid dog, and hence there is a higher chance for a transmission event to occur. In this study, we performed a sensitivity analysis on the dog traveling distances by scaling the mean traveling distances by factors 0.5 and 2. Our simulations demonstrated that longer travel lengths expand the distance that the disease may spread and result in higher numbers of cumulative cases and more dispersed distributions of cumulative rabies cases (S2 and S3 Figs).

In this study, we investigated three intervention strategies, i.e., dog population reduction, dog movement restriction, and mass vaccination. Despite dog density matters for rabies persistence, our results highlighted that reducing the dog population density could diminish the likelihood of rabies transmission. However, if the reduction proportion is inadequately high to clear susceptible dogs within the traveling length of rabid dogs, transmission is still feasible. For example, in Hatyai, where dog density is 243 dogs/km$^2$, culling half of the population still leaves 381 dogs within a one-kilometer roaming radius of a rabid dog. Our findings are also in line with several other research studies, which show that diminishing the dog population in a moderate way is unlikely to have a beneficial impact on rabies control [48–51]. This indicated that implementing only culling strategies might not be effective enough to eliminate rabies.

Owned dog confinement also has the potential to decrease the transmission likelihood. Owing to the dense aggregation area in Hatyai, limiting traveling distances of the owned dogs could cut more contacts among dogs than Tepha. We found that restricting owned dog

movement could effectively mitigate the transmission in both areas. Moreover, rabies could be eliminated if all owned dogs are restricted inside their homes. Although in the context of Thailand, in which most houses have no fences, strict enforcement on confinement of owned dogs might be impracticable. Therefore, this strategy might need to be implemented in combination with other interventions.

Our simulations also show that vaccination with high coverage might be a promising approach for achieving the goal of rabies elimination [4]. However, different levels of vaccination coverages, as well as vaccination strategies, are probably required to control rabies in different settings successfully. For instance, according to modeling studies conducted for dog populations in Africa, vaccination with 70% coverage annually is adequate to sustain herd immunity above a critical threshold [16,52]. However, based on our modeling results, vaccination coverage of 70% in Hatyai might still be insufficient.

A novel aspect of our study is the incorporation of the exact spatial distribution of buildings and roads in the individual-based rabies transmission model. The explicit representation of the geographical distribution of buildings and roads in our model allows us to explore the influence of geographical heterogeneity on rabies transmission. Since the difference in the geographical distribution of buildings and roads could affect the transmission dynamics of rabies, planning control measures might need to account for this effect. In densely populated areas, for example, a higher level of implementation may be required.

Like other studies, there were some limitations in our study. Firstly, we only consider the facilitator function of roads. Although our study sites are dominated by small local roads, some high-traffic roads could be geographical obstacles to dog roaming [25]. In addition, since dog movement pattern has not been exactly known, especially for rabid dogs, we assumed that they move randomly, and hence their spatial dispersal could be described by a Gaussian function. Roaming parameters were also assumed using a range of values from the literature. Third, as the exact dog population size was not available in Thailand, we estimated the dog population size based on the type and building density, the fraction of buildings with dogs, and the average number of dogs per building with dogs (Table 1). The owing fraction may vary from place to place. Lastly, although dog vaccination strategy was taken into consideration, dogs were vaccinated without regard for geographical priority. This may not reflect the real-world situation in which control measures are usually applied where a rabies outbreak appears.

## Supporting information

**S1 Text. Derivation of the encounter rate.**
(PDF)

**S1 Fig. Road alignments in (A) Hatyai and (B) Tepha**. In Thailand, roads are typically divided into 4 types, interregional highways, regional primary highways, regional secondary highways, and rural and local roads. Interregional highways are highways connecting Bangkok to outlying regions (for example, Route 4 to southern Thailand). Regional highways are highways within a region. Local roads are roads connecting main roads to important locations. The local roads are usually small roads where traffics is usually not heavy. Therefore, dogs can usually cross and walk along the roads. In the Hatyai area, there is 25.19 km of interregional highways, 15.12 km of regional primary highways, 53.74 km of regional secondary highways, and 2,708.52 km of local roads, representing 0.9%, 0.5%, 1.9%, and 96.7%, respectively. In Tepha, there are no interregional highways; the total length of regional highways and local roads is 65.08 km (5.4%) and 1,138.81 km (94.6%), respectively. The base layer of the map was obtained from https://data.humdata.org/dataset/thailand-administrative-boundaries.
(TIFF)

**S2 Fig. Effects of dog traveling distances.** We performed a sensitivity analysis on the dog traveling distances by scaling the mean traveling distances by factors 0.5 (x0.5) and 2 (x2). (A) Cumulative local cases within 365 days of simulations. (B) Likelihood for an imported infected dog to make a secondary infection. Blue and green represent Hatyai and Tepha, respectively. (TIFF)

**S3 Fig. Effects of dog traveling distances on the geotemporal pattern of rabies transmission.** We performed a sensitivity analysis on the dog traveling distances by scaling the mean traveling distances by factors 0.5 (x0.5) and 2 (x2). Each sub-figure depicts the spatial distribution of cumulative rabies cases after one year of rabid dog introduction. The greyscale represents the density of buildings (buildings/km$^2$), while the warm-color scale denotes the cumulative number of rabid dogs (dogs/km$^2$).
(TIFF)

**S4 Fig. The likelihood of a secondary infection caused by an imported rabid dog based on different proportions of G2 buildings owing unowned dogs.**
(TIFF)

**S5 Fig. The likelihood of a secondary infection caused by the importation of an imported rabid dog based on different proportions of G3 buildings owing unowned dogs.**
(TIFF)

**S6 Fig. The likelihood of a secondary infection caused by the importation of an imported rabid dog based on average number of unowned dogs per G2 buildings owing unowned dogs.**
(TIF)

**S7 Fig. The likelihood of a secondary infection caused by the importation of an imported rabid dog based on average number of unowned dogs per G3 buildings owing unowned dogs.**
(TIF)

**S1 Table. Locations of buildings in Hatyai.**
(XLSX)

**S2 Table. Locations of buildings in Tepha.**
(XLSX)

## Acknowledgments

We would like to thank the Department of Public Works and Town & Country Planning (DPT) and the Department of Livestock Development (DLD) for providing geographical maps and rabies data.

## Author Contributions

**Conceptualization:** Charin Modchang.

**Data curation:** Chayanin Sararat.

**Formal analysis:** Chayanin Sararat.

**Funding acquisition:** Anuwat Wiratsudakul.

**Investigation:** Chayanin Sararat.

**Methodology:** Chayanin Sararat, Suttikiat Changruenngam, Charin Modchang.

**Project administration:** Anuwat Wiratsudakul.

**Resources:** Arun Chumkaeo, Anuwat Wiratsudakul, Wirichada Pan-ngum.

**Software:** Chayanin Sararat, Suttikiat Changruenngam.

**Supervision:** Wirichada Pan-ngum, Charin Modchang.

**Validation:** Charin Modchang.

**Visualization:** Chayanin Sararat.

**Writing – original draft:** Chayanin Sararat, Charin Modchang.

**Writing – review & editing:** Suttikiat Changruenngam, Arun Chumkaeo, Anuwat Wiratsudakul, Wirichada Pan-ngum, Charin Modchang.

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
