## [Decision Letter · Decision Letter 0]

22 May 2021

Dear Dr. Modchang,

Thank you very much for submitting your manuscript "The effects of geographical distributions of buildings and roads on the spatiotemporal spread of canine rabies" for consideration at PLOS Neglected Tropical Diseases. As with all papers reviewed by the journal, your manuscript was reviewed by members of the editorial board and by several independent reviewers. In light of the reviews (below this email), we would like to invite the resubmission of a significantly-revised version that takes into account the reviewers' comments. 

The reviewers raise some significant issues that would need to be rectified, in particular regarding the lack of empirical data justifying the assumptions of the model

We cannot make any decision about publication until we have seen the revised manuscript and your response to the reviewers' comments. Your revised manuscript is also likely to be sent to reviewers for further evaluation.

Sincerely,

Daniel Leo Horton, PhD

Associate Editor

Sergio Recuenco

Deputy Editor

The reviewers raise some significant issues that would need to be rectified, in particular regarding the lack of empirical data justifying the assumptions of the model

Reviewer's Responses to Questions

**Key Review Criteria Required for Acceptance?**

**Methods**

-Are the objectives of the study clearly articulated with a clear testable hypothesis stated?

-Is the study design appropriate to address the stated objectives?

-Is the population clearly described and appropriate for the hypothesis being tested?

-Is the sample size sufficient to ensure adequate power to address the hypothesis being tested?

-Were correct statistical analysis used to support conclusions?

-Are there concerns about ethical or regulatory requirements being met?

Reviewer #1: The authors present a manuscript on the effects of geographical distributions of buildings and roads on the spatiotemporal spread of canine rabies. The authors construct an individual based model or rabies transmission the explicitly considers the built environment and test also non-pharmaceutical interventions like density reduction and confinement of dogs. Although the approach is interesting, the paper is of limited scientific value because of the following reasons:

1. The model is almost exclusively based on assumptions and not on empirical observations (Table 1). 

2. There are several issues with the the model assumptions as outlined below:

All of the buildings in the city are classified into 3 kinds: (G1) houses/residential, (G2) public places where stray dogs usually can be found, (G3) other types of buildings (e.g., groceries, hotels, banks, etc.). Distance of each building centroid to the closest road is estimated approximately.

Assumption of dog locations: Owned dogs in G1, 90% of stray dogs in G2, 10% of stray dogs in G3

* There is no empirical basis shown for such distribution, especially the 90-10 division.

Main component of the model is the encounter rate expression, which is simply the distance between two spherical Gaussians. There are several points to note about the implicit assumptions:

* Such movement is often not Gaussian because of the limitations of man-made structures: the distribution stretches widely along the roads while being very narrow around the buildings. 

* Even if a Gaussian could be fitted for simplicity reasons, that would not be a spherical one, the whole covariance matrix (2x2) must be stated in order to account for the stretching of the distribution. At that point, the derived expression in the paper does not apply.

* The variances of these distributions (i.e. squared mean traveling distances) are somewhat naively determined: some of the values are taken relatively to the data at hand (e.g. lines 171-173), and nearly all of the values are at the end noted as assumptions in Table 1.

3. There is a fundamental shortfall in the consideration of scaling of rabies transmission. We know that dog rabies transmission depends on direct dog to dog transmission on scales of tens of kilometers but also human mediated dog transport on scales of hundreds of kilometers. This interaction is complex and transmission in urban centres often remains endemic only because of continuous importation of rabies, which drive the transmission much more than small scale (< 10 km) spatial effects of the built environment. Unless the authors provide empirical data on the interplay of local transmission between dogs and human mediated dog transport, the analysis of the built environment remains meaningless.

Reviewer #2: (No Response)

**Results**

-Does the analysis presented match the analysis plan?

-Are the results clearly and completely presented?

-Are the figures (Tables, Images) of sufficient quality for clarity?

Reviewer #1: The main issue is with the results: there is a circular logic as they design to model to contain relative building-to-road distance as a parameter, and they report that the simulations show different results for cities with different average building-to-road distances. Furthermore, such simulations, considering the number of hyperparameters (i.e. assumptions) can be very misleading since it is fairly easy to find a set of parameters that show somewhat similar distribution (inside given confidence intervals, as in Fig.4 - a) with the data that consists of a single sample. Yet, if the exact relationship of the distribution and these assumptions cannot be explained, then there is no way to say which ones were the main contributing factors and which ones were trivial.

As a note, the last sentence is why likelihood maximization over a fixed model is much better than simulation studies, since the factors that contribute to the likelihood are much easier to distinguish than in simulations.

The handling of roads seems to be essentially as a passage way for dogs. Hoewever from our experience this is highly dependent on traffic. While roads are unambiguous passage ways in low traffic community areas, high traffic roads are important geographical barriers almost interrupting dog-to-dog contact networks1. This is an important shortfall of this paper.

1. Laager, M. et al. The importance of dog population contact network structures in rabies transmission. PLoS neglected tropical diseases 12, e0006680, doi:10.1371/journal.pntd.0006680 (2018).

Reviewer #2: (No Response)

**Conclusions**

-Are the conclusions supported by the data presented?

-Are the limitations of analysis clearly described?

-Do the authors discuss how these data can be helpful to advance our understanding of the topic under study?

-Is public health relevance addressed?

Reviewer #1: The conclusion is a bit disappointing for such an extensive modelling exercise. 

We know since a long time that the reduction of dog density and that confined dogs reduce transmission. This is not new. The reduction of dog density is not a feasible elimination strategy and to confine dogs effectively requires a very high social control. The argument of the authors that the consideration of the distribution of buildings could contribute to target control strategies is not valid. The essential point of effective interventions like mass vaccination of dogs is to reach a sufficiently high coverage over a whole area irrespective of the spatial heterogeneity of the built environment.

Reviewer #2: (No Response)

**Editorial and Data Presentation Modifications?**

Reviewer #1: not applicable

Reviewer #2: (No Response)

**Summary and General Comments**

Reviewer #1: Altogether this paper does not contribute to new knowledge because most of the model parameters are based on assumptions and not on empirical data. The proposal of dog population reduction and total confinement are not feasible. The WHO recommendation of mass vaccination is not addressed by the model. This would have shown that the essential driver of elimination is reaching sufficiently high vaccination coverage, the so called herd immunity.

Reviewer #2: The manuscript describes an epidemiological simulation model for rabies spread in dog populations in two provinces in Thailand. The model is based on the geographical distribution of buildings and roads in the two locations and found that the structure influences rabies spread. The authors used the model to simulate the effect of dog density reduction and confinement of owned dogs. The manuscript is well written and can be (mostly) easily followed. Also, it is innovative in the sense that geographical feature of an area is investigated in how this influences rabies spread.

However, there are some points that should be improved before the article can be published.

Main comments:

1. The authors used an approach for rabies incursion into a fully susceptible population (1 dog infectious, all other susceptible). This most likely does not reflect the reality in Thailand, where rabies is endemic since a long time. Most likely, a good proportion of the dog population is not susceptible. It has been mentioned as a limitation of the study that neither vaccination nor other immediate outbreak response were considered in the model. However, the discussion on that aspect needs to be extended. What would be the effect if vaccination were considered? Most likely, this would not be evenly distributed amongst the dog population, but be higher in the owned dogs. Ideally, vaccination is included into the model. This can be done in a simple way by assuming that at the start of the epidemic, x% of the owned dog population is vaccinated. 

2. Related to the former comment, I suggest to extend in the introduction recommended and available rabies control interventions. In my opinion, it needs to be said that vaccination of dogs it the most promising strategy to control the disease. It can be argued that in addition, other control options should also be considered, such as dog population reduction and dog confinement. This is what the authors did in this work.

3. The authors state that because the dog population in these two districts is unknown, they derive from the distribution of buildings how many owned and stray dogs are located in the study sites. This results in a dogs population of mainly owned dogs and very few stray dogs (compared to the owned dogs). The number of dogs in the study sites should be reported earlier than at line 314, ideally at the position where the number of buildings are given. A rough calculations shows that in Hatyai there are 24 times more owned that stray dogs, and in Tepha there are 68 times more owned dogs. Is this a realistic scenario? Also, the distribution of the dog population is random, considering the density of the different types of buildings (e.g. always 54% of the G1 buildings are occupied by dogs). Is this assumption also realistic? Aren’t there clusters of dogs in the study site, where this proportion may be larger or lower? The same for the distribution of stray dogs.

4. Distribution and behavior of stray dogs: It is assumed that the stray dogs are located at a given building and that they stay there (as pet dogs stay at their houses). Is this realistic? Also, do stray dogs roam similarly than owned dogs? I would have expected that they roam further.

5. The large majority of model parameters are based on assumptions. I was wondering whether there are any possibilities from the literature on which the authors could base their assumptions. There are a range of articles on the roaming behavior of free-roaming dogs from which the authors may take information on the distance that dogs travel. Also, it looks strange to me that the average number of dogs per dog-owning household is an assumption (2.67). How did the authors assume that number? Also the birth rate is very specific, how was that value assumed? For the transmission probability is it said "estimation". How did the authors estimate this value? Also, it looks to me very low, as the probability that rabies is transmitted through a bite of a rabid animal is 0.49 according to ref [39]. Finally, I highly recommend to do a sensitivity analysis for these parameters to understand how the model reacts towards the change of the parameter values.

6. One of the main findings is, according to the authors, that the dog density is essential for rabies spread. However, they also found that the reducing the dog population density does not seem to greatly influence the probability of making a secondary transmission after importation of a rabid dog. In addition, it is propagated in the literature that rabies is not density dependent. Finally, it was not clear to me whether the population reduction only considers owned or owned and stray dogs. Is the reduction been done randomly? I suggest that the authors more carefully discuss the (notably very interesting) finding of the dog density for rabies spread. 

7. Dog confinement: as for dog population reduction, I also find this aspect worthwhile to be investigated. But I have some questions: Why are confined dogs still able to roam 100m on average? This still allows them to contact other dogs Is this assumption correct? Is it also a realistic assumption that almost 50% of the owned dogs are confined? This again raise the issue that a sensitivity analysis of the model is very important to see the influence of the parameters on the model outcomes. Finally, it is not that surprising that 100% confinement of the dogs allow to eliminate a rabies outbreak. However, it should be more discussed that even though the confinement is not 100% (because they can still roam 100m on average) and even though stray dogs are around and not confined (if I understood this correctly), there is still a promising effect of the rabies control by this strategy. This, together with the practicability in Thailand to undertake confinement of all owned dogs, has to be better discussed.

8. Only one type of road is considered in the model. I would expect that the size of road would make a difference. There may be roads that helps the dogs to roam (as considered in the here described model), but there may also be roads that are clear barriers for roaming (as e.g. shown in Laager et al (doi: 10.1371/journal.pntd.0006680)). In addition, the direction of the road is important for the direction of the pronounced roaming. I do not think that this all needs to be implemented in the model (as this is a complex procedure), but it should at least be discussed. 

9. Writing style: There are some passages in all sections (introduction, M&M, results and discussion) that I would transfer to another section. E.g. line 108-121 better fits to the M&Ms; and at the end of the introduction it is more needed to state the aim of the study and its potential impact, rather than describing the model already here. Line 221-223 better fits to M&M. In general, the page 12 reads a bit as a mix of results and M&Ms. 

Minor comments

1. Why did the authors took the probability of a secondary transmission as the outcome metric to investigate the effect of control actions, and not the number of rabies cases after a certain time of rabies propagation?

2. The number of simulated cases in Hatyai indeed fits quite well with the reported data. It has been said that for Tepha it is not possible to undertake such a comparison, because no rabies cases are reported. However, it may also be argued that the number of cases is zero, and then it is possible to make a comparison with the conclusion that the model over-predicts the number of rabies cases. Also, what is the source of the number of cases in Hatyai.

3. Where are the randomly imported dogs located? Is this always an owned dog, or also stray? I think for both types of dogs an importation is possible.

4. When the term "significant" is used, please support it by a statistical test.

5. Line 83-84: I suggest to edit: "… reservoir for rabies, rabies transmission to humans by dog bites may anytime happen, and the number…"

6. Line 92 (ref 15-18): only one of these references is related to dog rabies, all other are investigation wildlife rabies. This should be made clear, as wildlife are differently distributed than dogs, which depend on humans. 

7. Line 136: Please delete "where stray dogs usually can be found" because this comes later and raises question about the role of the other buildings.

8. Line 148-149: please better describe the distributions of the stray dogs in the different buildings, as the number of dogs in G2 and G3 differs. This can only be found in table 1, thus it is important to refer to the table at this position.

9. Line 154: Please delete "in this work, "

10. Line 165: I suggest to use another symbol than d for the death rate, so that it cannot be mixed up with the distance d. Please describe that the death rate was selected as being the size to keep the population constant (I guess).

11. Figure 1b: please include the death rate into the model scheme.

12. Line 167: note that Kij is the encounter rate.

13. Line 210: I suggest to write "… the model classifies each single dog into susceptible, …"

14. Line 237-238: The authors should explain what they mean with "different order of closeness". 

15. Line 248-249: can be deleted, is a repetition. 

16. Line 271-273: can be deleted (repetition)

17. Line 348: rabid instead of rapid

18. Line 348: these references only consider wildlife, not dog rabies

19. Lines 351-363: this is repetition and could be deleted

20. Line 369: please delete "typically"

21. Line 370: put a comma after "dogs" and delete "so"

22. Line 401: it is said that the geographical setting matters for rabies spread. On the other hand, one may also argue that the conclusions from the study in regards to control measures tested are the same in both regions. Thus the difference in the geographical setting between the two study sites tested here does not result in different conclusions. 

23. The limitation of not considering vaccination nor "immediate response" is mentioned here. But it is not discussed what may be the effect of this non-consideration. Also, I was wondering whether the author could calculate the average reproductive ratios from their model to have an idea how much this corresponds with what is expected.

PLOS authors have the option to publish the peer review history of their article (what does this mean?). If published, this will include your full peer review and any attached files.

Reviewer #1: No

Reviewer #2: No
---

## [Decision Letter · Decision Letter 1]

11 Nov 2021

Dear Dr. Modchang,

Thank you very much for submitting your manuscript "The effects of geographical distributions of buildings and roads on the spatiotemporal spread of canine rabies" for consideration at PLOS Neglected Tropical Diseases. As with all papers reviewed by the journal, your manuscript was reviewed by members of the editorial board and by several independent reviewers. In light of the reviews (below this email), we would like to invite the resubmission of a significantly-revised version that takes into account the reviewers' comments. 

Thank you for your revised manuscript which has addressed many of the reviewer's comments. Please address those that remain below.

In particular, the issue of lack of empirical data supporting the assumptions remains.

 I suggest that an explanation for the assumptions around dog movement distances are clearly included in the methods, and parameter table 1 (supported where appropriate by empirical data from other studies referenced in lines 402-414, and reference 35 and local information to put them into regional context). 

The sensitivity analyses are a worthwhile addition demonstrating a large effect on transmission. The reader will be interested in their effect on the main conclusions (Figure 6). If uncertainty over dog movements has a significant impact on the relative success of interventions then that is an important conclusion in itself.

I also recommend using 'free-roaming' rather than 'stray'

We cannot make any decision about publication until we have seen the revised manuscript and your response to the reviewers' comments. Your revised manuscript is also likely to be sent to reviewers for further evaluation.

Sincerely,

Daniel Leo Horton, PhD

Associate Editor

Sergio Recuenco

Deputy Editor

Thank you for your revised manuscript which has addressed many of the reviewer's comments. Please address those that remain below.

In particular, the issue of lack of empirical data supporting the assumptions remains.

 I suggest that an explanation for the assumptions around dog movement distances are clearly included in the methods, and parameter table 1 (supported where appropriate by empirical data from other studies referenced in lines 402-414, and reference 35 and local information to put them into regional context). 

The sensitivity analyses are a worthwhile addition demonstrating a large effect on transmission. The reader will be interested in their effect on the main conclusions (Figure 6). If uncertainty over dog movements has a significant impact on the relative success of interventions then that is an important conclusion

I also recommend using 'free-roaming' rather than 'stray'

Reviewer's Responses to Questions

**Key Review Criteria Required for Acceptance?**

**Methods**

-Are the objectives of the study clearly articulated with a clear testable hypothesis stated?

-Is the study design appropriate to address the stated objectives?

-Is the population clearly described and appropriate for the hypothesis being tested?

-Is the sample size sufficient to ensure adequate power to address the hypothesis being tested?

-Were correct statistical analysis used to support conclusions?

-Are there concerns about ethical or regulatory requirements being met?

Reviewer #1: The authors have made a massive effort in revising this paper. However, I have still a major reservation because of the large number of assumptions the authors still make. I recommend the authors to collect more primary data to substantiate their assumptions. In my own work my assumptions were often wrong and we must first observe nature and understand it deeply prior to attempting to model it.

The authors maintain an importation parameter in their model that reflects the human mediated dog transport. The claim that this is estimated based on incidence data. But how can they distinguish between the incidence provided from local chains of transmission and the importation from the outside? It would make a lot of sense in this model to collect empirical dog importation data first.

The authors explain the different types of roads in the study area, but we still don't know which of these roads are facilitating dog movement and which ones are barriers for dog movement. Again more empirical data on this would make the models much more realistic.

Reviewer #2: please see "Summary and General Comments"

**Results**

-Does the analysis presented match the analysis plan?

-Are the results clearly and completely presented?

-Are the figures (Tables, Images) of sufficient quality for clarity?

Reviewer #1: I still feel at unease with this paper because the results still build on a large number of assumptions of which we don't know if they hold true. Again, I recommend the authors to collect first more empirical data on dog ecology in Thailand prior to develop a mathematical model.

Reviewer #2: please see "Summary and General Comments"

**Conclusions**

-Are the conclusions supported by the data presented?

-Are the limitations of analysis clearly described?

-Do the authors discuss how these data can be helpful to advance our understanding of the topic under study?

-Is public health relevance addressed?

Reviewer #1: The authors have modified the conclusion by addressing dog mass vaccination as most important recommended strategy. This is however not a primary conclusion of this paper and hence the reader does not understand what is the ultimate purpose of this model. Although I recognize the hard work of revising this paper, I still feel that the knowledge gain is too small because of the high uncertainty with all the assumptions.

Reviewer #2: please see "Summary and General Comments"

**Editorial and Data Presentation Modifications?**

Reviewer #1: The data presentation is acceptable.

Reviewer #2: please see "Summary and General Comments"

**Summary and General Comments**

Reviewer #1: See my points under Conclusion

Reviewer #2: Thank you for the great revision of the manuscript. Most of the concerns and uncertainties were cleared, however there are still a range of comments and open questions:

General comment: I think it needs to be made clear that the link between geographical situation and rabies spread goes via the dog population simulated in the model. There the crucial assumption taken in the model that a high building and road density leads to a larger number of dogs and to more roaming, respectively. This needs to be indicated clearly from the start (also in the abstract). This is particularly important because the interventions are taken on the dogs, not on the buildings and roads. So it needs to be understood that dog population is explicitly modeled here.

Line 36: was vaccination also only done in owned dogs?

Line 37: Please explain how can these results direct policy.

Line 94: Did this increase in the number of rabies cases happen without any change in the surveillance system?

Line 120: can "wild animals" be replaced by raccoons?

Line 123: "…and alignments of roads." Suggest to add here " and use them to simulate the dog population in these area." Or similar.

Line 135: How is the reporting of rabies cases done? What type of surveillance does exist in this region? This may give an idea of underreporting.

Line 137: It is great that you assessed the quality of roads, but it should be described somewhere in the methods (probably in this section). I am however not sure if the correct thresholds are taken. I agree that the max 5% of large roads can be ignored, but I guess that there is also variation in the here called "local roads" in terms of building barriers for dogs. Such assumption that can be better discussed. 

Line 141: How was this categorization of buildings done? Visually by looking at the maps, or otherwise?

Line 145: Please also state for the G3 buildings how they host dogs.

Line 162 (or close by): There is a sentence missing saying that a proportion of owned dogs are modelled to not roam outside (if this is correct).

Line 164: delete "then"

Line 165: I suggest to never use "see" before you refer to a table or figure. This is uncommon in scientific papers.

Line 140: Please state that because of this feeding, they can be assumed to stay around a building that can be classified as their home building (if correct). 

Lin 168: what type of building? Only G2 or also G3?

Line 170:. I would say that these are not estimations, but assumptions according to table 1. Also, I suggest to report the exact percentage of stray dogs in G2 and G3 buildings based on the assumptions (if I am correct almost 100x more in G2 buildings) already here. 

Line 192: please describe what type of dogs (owned or unowned?) are considered to immigrate and discuss that also the other type can immigrate and think about what effect this might have to not consider the other type of dogs.

Line 200: transmission rate p: indicate that this is probability of bite * probability of transmission given a bite. It is said "model fitting": please explain how this is done. Was the fit done to the rabies observation data? Is so (which is a typical approach by model building), it is expected that the model predictions correlate with the observed data. So this argument cannot be taken as a criteria how good the model represents reality, but only that it well reproduces what is observed. (see also my comment further down for line 384). If p = 1*10-5, then the probability of a bite happening when two dogs encounter is very low, i.e. roughly 2*10-5, e.g. only two times per 10'000 occasions. Is this realistic? Or do I do the maths wrong?

Table 1: please replace public places by G2 to be consistent

Line 222: This distance of 1km a bit comes out of the blue. Why did you pick this number? This is the first time this number if mentioned. I would have expected to describe this already in the population and movement section.

Line 246 ff: I do not understand this sentence. Also I think it would better fit to the methods section. 

Fig 3: please add the names of the districts in the graphs a) and b). Also, b) is 10times more dense than a) Is this correct?

Line 302: Please quickly provide the conclusion what was found in the sensitivity analysis here.

Line 333: I suggest to stick to % and not use the decimal for 0.8.

Line 343: 0.33 to 0.78: this is an increase, not a reduction. 

Fig 6: why are the starting points of the likelihood different for each simulation?

Line 365: I suggest to use "small scale structures" rather than "exact" because for the larger scale models the structures may also have been included exactly. 

Line 384: which is expected because the incidence data were used to fit the model parameters.

Line 402: I suggest to delete "however"

Line 408: Based in the data provided from Chad, why do you come up with 1km max? There is another study (doi: 10.3389/fvets.2021.617900) that would better support your distances.

Line 424: The comparison with the study in Tanzania might be critical, because I guess when you reduce the population in your model by 100, there might be almost no transmission anymore? Also, I am wondering whether these dogs living in lower densities do not roam further. 

Line 449: not only modelling studies have limitations, but in all sorts of studies.

Line 455: "especially for rabid dogs…" This opens a substantial of follow-up question: you considered healthy dogs in the model, how much do the results hold true when dogs become rabid?

Line 458: and what is the consequence of this? Do you think this estimation is appropriate? Can it be compared with other studies in Thailand?

I think REF 33 and 37 are the same.

PLOS authors have the option to publish the peer review history of their article (what does this mean?). If published, this will include your full peer review and any attached files.

Reviewer #1: No

Reviewer #2: No
---

## [Decision Letter · Decision Letter 2]

6 Apr 2022

Dear Dr. Modchang,

We are pleased to inform you that your manuscript 'The effects of geographical distributions of buildings and roads on the spatiotemporal spread of canine rabies: An individual-based modeling study' has been provisionally accepted for publication in PLOS Neglected Tropical Diseases.

Best regards,

Sergio Recuenco

Deputy Editor

Sergio Recuenco

Deputy Editor

Reviewer's Responses to Questions

**Key Review Criteria Required for Acceptance?**

**Methods**

-Are the objectives of the study clearly articulated with a clear testable hypothesis stated?

-Is the study design appropriate to address the stated objectives?

-Is the population clearly described and appropriate for the hypothesis being tested?

-Is the sample size sufficient to ensure adequate power to address the hypothesis being tested?

-Were correct statistical analysis used to support conclusions?

-Are there concerns about ethical or regulatory requirements being met?

Reviewer #1: The authors have further improved the paper by including a senstivity analysis. There remain critical questions to answer.

Table 1: The authors assume that the mortality is equal to the birth rate. On which grounds do they make this assumption? Is there human dog consumption in the study area?

All values dog travelling are assumptions which should be substantiated from the existing literature. Particlularly the travelling distance of dogs living in building near roads without confineable fences of 5 km is from our experience to big. The authors should explain on which empirical grounds they make these assumptions.

With regard to dog importations, I don’t agree that two successive locally transmitted rabies cases are unlikely to be separated longer than 25 days. On average a locally exposed dog will become rabid within a month but this can clearly also be longer. At least the authors clarify their assumptions.

**Results**

-Does the analysis presented match the analysis plan?

-Are the results clearly and completely presented?

-Are the figures (Tables, Images) of sufficient quality for clarity?

Reviewer #1: (No Response)

**Conclusions**

-Are the conclusions supported by the data presented?

-Are the limitations of analysis clearly described?

-Do the authors discuss how these data can be helpful to advance our understanding of the topic under study?

-Is public health relevance addressed?

Reviewer #1: Altogether, I still see this manuscript as an exercise in individual based modelling of the role of buildings and roads applied to rabies which must be validated with empirical data, otherwise the knowledge gain for rabies control is very limited. Given the socio-economic status of Thailand, it is conceivable that a policy of compulsory dog rabies vaccination would be feasible. This is a point the authors should address as a scenario in relation to the expected overall coverage of the owned and ownerless dog population. In this way the model could be useful for prospective rabies elimination in Thailand.

**Editorial and Data Presentation Modifications?**

Reviewer #1: (No Response)

**Summary and General Comments**

Reviewer #1: (No Response)

PLOS authors have the option to publish the peer review history of their article (what does this mean?). If published, this will include your full peer review and any attached files.

Reviewer #1: **Yes: **Jakob Zinsstag

---

## [Editor Report · Acceptance letter]

6 May 2022

Dear Dr. Modchang,

We are delighted to inform you that your manuscript, "The effects of geographical distributions of buildings and roads on the spatiotemporal spread of canine rabies: An individual-based modeling study," has been formally accepted for publication in PLOS Neglected Tropical Diseases.

Best regards,

Shaden Kamhawi

co-Editor-in-Chief

Paul Brindley

co-Editor-in-Chief
